# Novel CF₃-Substituted Pyridine- and Pyrimidine-Based Fluorescent Probes for Lipid Droplet Bioimaging

**DOI:** 10.3390/ijms26115271

**Published:** 2025-05-30

**Authors:** Dmitrii L. Chizhov, Yuriy A. Kvashnin, Nadezhda S. Demina, Ekaterina F. Zhilina, Artem S. Minin, Natalia A. Verbitskaia, Ekaterina M. Dinastiia, Gennady L. Rusinov, Egor V. Verbitskiy, Valery N. Charushin

**Affiliations:** 1I. Ya. Postovsky Institute of Organic Synthesis, Ural Branch of the Russian Academy of Sciences, S. Kovalevskaya Str., 22, 620066 Ekaterinburg, Russia; dlchizhov@ios.uran.ru (D.L.C.);; 2Institute of Natural Sciences and Mathematics, Ural Federal University, Kuibysheva St. 48, 620026 Ekaterinburg, Russia; 3Department of Organic and Biomolecular Chemistry, Ural Federal University, Mira St. 19, 620002 Ekaterinburg, Russia

**Keywords:** pyridines, pyrimidines, aggregation-induced emission, lipid droplets, fluorescent probes

## Abstract

We have designed novel push–pull systems based on CF_3_-substituted pyridines and pyrimidines. The photophysical properties of these new fluorophores have been examined using both absorption and emission spectral analyses in acetonitrile solutions and solid states. All fluorophores proved to exhibit moderate absolute quantum yields of up to 0.33 in solutions and up to 0.12 in solid states, depending on their specific structures. Most fluorophores have demonstrated significant aggregation-induced emission behavior, making them suitable as robust and low-toxicity bioimaging agents for bioimaging studies. Comparison with known dyes and studies on various cell cultures demonstrated the selectivity of the obtained push–pull systems for visualizing lipid droplets.

## 1. Introduction

Lipid droplets are cell organelles that accumulate lipids and consist of a phospholipid monolayer and a neutral lipid core (cholesterol ester and triglyceride) [1]. The formation of lipid droplets takes place in the endoplasmic reticulum as a highly dynamic process [2]. Lipid droplets play one of the crucial roles in maintaining the regular life activities of cells [3].

A number of human body disorders, such as obesity [4], non-alcoholic fatty liver disease [5,6], and atherosclerosis [7] are considered to be related to the accumulation of lipid droplets. Therefore, prompt and opportune checking of the alterations connected with lipid droplets is very important for realizing the cell physiological processes, as well as the elaboration of appropriate schemes for the treatment of diseases. Fluorescent imaging targeted at monitoring the biological functions of lipid droplets has demonstrated great potential during the recent years because of its real-time application, superior sensitivity, and high signal-to-noise ratio [8,9,10,11,12,13]. Furthermore, fluorophores of this kind possess an aggregation-caused quenching effect at high staining concentrations, which causes inappropriate bioimaging quality with a low signal-to-noise ratio. These disadvantages can be overcome by using aggregation-induced emission (AIE) fluorescent probes with large Stokes shifts, high fluorescence, and good biocompatibility [14].

Currently, aggregation-induced emission fluorescent probes with push–pull structures are inspiring candidates for lipid droplet imaging due to the easy tuning of their optical properties and good lipophilicity [14,15,16,17]. It should be noted that incorporation of the CF_3_ group in biologically active agents also contributes to enhancing their lipophilic characteristics without metabolic vulnerability [18].

Having a significant background in the synthesis of push–pull systems [19,20,21,22] for material science applications, we have designed and synthesized two novel series of push–pull systems based on CF_3_-substituted pyridines (**I**) and pyrimidines (**II**) to study their photophysical properties and plausible applications as fluorescent probes towards lipid droplets (Figure 1).

## 2. Results and Discussion

### 2.1. Synthesis of Push–Pull Systems

The parent chloro-derivatives of pyridine **5** and pyrimidine **7** used in this work were synthesized from 4-acetyl-*N,N*-dimethylaniline **1** through multistep procedures (see Figure 1). The 4-substituted acetophenone 1 was converted into diketone **3** by exploiting the Claisen condensation with ethyl trifluoroacetate 2 in the presence of LiH. Despite the fact, that the reaction of 1,3-diketones with urea underlies a classical 2-hydroxypyrimidines synthesis, our attempts to obtain 2-hydroxypyrimidine **6** through the reaction of diketone **3** with urea under classical conditions (boiling ethanol, acetic acid) were unsuccessful. Previously, it has been shown that the rate and regioselectivity of heterocyclizations of fluorinated 1,3-diketones with several *N*,*N*-binucleophiles can be enhanced in the presence of triethyl borate [23,24]. Therefore, we have decided to use the latter reagent in this reaction. In fact, the addition of a considerable excess of triethyl borate (4 equiv.) ensured the complete conversion of the reactants in refluxing acetonitrile for 20 h to furnish 6 in 88% yield (isolated). The reaction of diketone **3** with 2-cyanoacetamide proved to proceed in refluxing pyridine for 20 h, thus giving 2-hydroxy-substituted pyridine **4** in 87% yield.

The next step of dehydroxychlorination of **4** and **6** proceeded smoothly in refluxing POCl_3_ in the presence of tetraethylammonium chloride to give the corresponding chlorides **5** and **7** in 83 and 79% yields, respectively. Noteworthy, the yields did not exceed 50–55% when other sources of chloride ions (triethylammonium or pyridinium hydrochloride) were used.

To prepare the target 6-[4-(dimethylamino)phenyl]-2-(het)aryl-4-(trifluoromethyl) nicotinonitriles (**9a**–**e**) and [2-(het)aryl-6-(trifluoromethyl)pyrimidin-4-yl]-*N*,*N*-dimethylanilines (**10a**–**e**), the previously [25] developed Suzuki cross-coupling procedure has been used. It was based on the interaction of the parent compounds **5** or **7** with the corresponding pinacol esters of 4-(diphenylamino)phenylboronic (**8a**), 9*H*-carbazole-9-(4-phenyl)boronic (**8b**), 9-ethyl-9*H*-carbazole-3-boronic (**8c**) acids, (4-fluorophenyl)boronic acid (**8d**), or (4-(trifluoromethyl) phenyl)boronic acid (**8e**), proceeding in refluxing 1,4-dioxane in the presence of K_3_PO_4_ and Pd(PPh_3_)_4_ as catalyst (Figure 2). The identity and purity of push–pull systems **9** and **10** were confirmed by ^1^H, ^19^F, and ^13^C NMR spectroscopy, as well as elemental or HRMS analysis, respectively. All the products prepared were proved to have satisfactory analytical data (Appendix A).

### 2.2. Photophysical Studies of the Obtained Fluorophores ***9a***–***e*** and ***10a***–***e*** in Solutions and Solid States

The photophysical properties of the synthesized compounds **9a**–**e** and **10a**–**e** at room temperature were investigated by applying UV-Vis and photoluminescence spectroscopy in acetonitrile solutions and solid state (Table 1, Figure 2, Figure 3, Figure 4, Figure 5, Figure 6, Figure 7, Figure 8, Figure 9, Figure 10, Figure 11 and Appendix A). The main experimental data are summarized in Table 1.

All obtained compounds have similar patterns of optical properties. Two main regions can be distinguished in the absorption spectra with maxima in the range of 200–300 nm (ε ~13,000–43,000 M^−1^·cm^−1^) and 340–420 nm (ε ~30,000–56,000 M^−1^·cm^−1^). Absorption peaks in the short wavelength range correspond to local allowed π–π* transitions. In contrast, absorption in the long wavelength range can be attributed to intramolecular charge transfer or have a mixed character. The presence of the electron-accepting CN-group in **9a**–**e** (λ_abs_ 403–413 nm) results in a bathochromic shift of the maximum of the long-wavelength absorption band compared to **10a**–**e** (λ_abs_ 372–381 nm).

The emission of solutions of compounds **9a**–**e** and **10a**–**e** in MeCN is observed in the range of 490–620 nm. The increase in the electron-withdrawing character of the substituents upon transition from CF_3_– to F–substituted derivatives (**9d**, **10d** vs. **9e**, **10e**), as well as the transition from triphenylamino-substituted to the more rigid 9-phenyl-9*H*-carbazole derivatives (**9a**, **10a** vs. **9b**, **10b**), leads to pronounced bathochromic shifts of the emission maximum by 5–27 nm. The fluorescence quantum yields of the compounds studied are generally varied from low to moderate ones. The highest values in each series, **9a**–**e** and **10a**–**e**, are found in compounds **9c** (0.11) and **10c** (0.33), which bear a 9-ethyl-9*H*-carbazolyl substituent (Table 1). For both of these compounds, there is an order of magnitude increase in the irradiative transition rate constant (*k*_r_) values, as shown in Appendix A. The decomposition kinetics and lifetime values for compounds **9a**–**e** and **10a**–**e** were described by bi-exponential dependences. The lifetimes evaluated for **9a**–**e** and **10e** are shorter than those for **10a**–**d**, which is probably due to an increasing nonradiative relaxation through vibrational modes (Appendix A).

It should be emphasized that in the solid state, the blue-shifted (Δλ = 20–63 nm) emission spectra maxima (430–540 nm) were observed versus the same solutions for all compounds **9a**–**e** and **10a**–**e** (Appendix A vs. Appendix A). This fact can be explained by the formation of a twisted intramolecular charge transfer (TICT) state in MeCN and a locally excited state in the solid state due to restricted intramolecular rotation, which causes blue-shifted emission in the solid state [26]. The particularly large Stokes shifts (>10,000 cm^−1^ for **10e**) could similarly indicate the presence of a TICT excited state (Table 1). The fluorescence quantum yields of the compounds **9a**–**e** and **10a**–**e** in the solid state were low to moderate. The presence of three components in the description of fluorescence decay of compounds **9a**–**e** and **10a**–**e** indicates that fluorophores can exist under three conformational states (Appendix A). The excitation and emission spectra of these three fluorescent states of **9a**–**e** and **10a**–**e** are highly overlapping and inseparable at room temperature. Therefore, the transitions between them can be resolved using fluorescent lifetime measurements.

The powders of the compounds ranged from off-white and yellow for **9a**–**d** and **10a,c**–**e** to orange and red-colored for **10b** and **9e** in daylight (Figure 2, Figure 3 and Figure 4). On the contrary, the same powders exhibited an intense emission from the green-yellow for compounds **9a**–**e** and **10a**–**d** to orange for compound **10e** (Figure 4). The emission in the solid state for compounds **9a**–**e** was the same, namely green, as recorded in solutions.

The emission for compounds **9a**–**e** in the solid state was yellow/green, consistent with the recorded emissions in solution.

In addition, the aggregation-induced emission (AIE) properties of each compound series **9a**–**e** and **10a**–**e** have been estimated. To evaluate the AIE properties of compounds, their emission spectra were recorded in a mixture of MeCN and water with various parts of the latter (Figure 5, Figure 6, Figure 7, Figure 8 and Appendix A). AIE properties were observed in compounds **9a**, **9b**, **9e**, **10a**, **10b**, and **10e**, resulting in an increase in fluorescence intensity from 1.6 to 80 times.

The fluorescence spectra of **9** and **10** in MeCN/H_2_O mixtures with various water fractions (*f*_w_) at the appropriate excitation wavelength are shown in Figure 5, Figure 6, Figure 7, Figure 8 and Appendix A. The best results were observed for CF_3_-substituted derivatives **9e** and **10e**. For both compounds **9e** and **10e**, quite weak emission peaks were demonstrated in pure MeCN, while a dramatic increase in the emission intensity was observed when *f*_w_ increased from 70 to 85%. At the same time, the fluorescence intensity exhibited an increasing trend of up to 80 times enhancement at *f*_w_ = 95% for compound **10e** (Figure 8), inducing new bands, centered at 532 and 476 nm for **9e** and **10e**, respectively. Absolute quantum yields were 0.19 and 0.08 for **9e** and **10e** aggregates, respectively, which are much higher than those for the same substances in the solid state. We suggest that a restriction of the intramolecular motion of substituents can be one of the main possible mechanisms for the observed enhancement of fluorescence intensity [27]. Additionally, the blue-shifted emission observed in aggregates and in the solid state can also be linked to restricted intramolecular rotation.

### 2.3. The Cytotoxicity, Cellular Uptake, and Fluorescence Imaging of Compounds **9a**–**e** and **10a**–**e**

Before conducting cellular experiments with probes **9a**–**e** and **10a**–**e**, we established a Resazurin cell viability assay using live *Vero* cells (green monkey kidney epithelial cell culture) to evaluate cytotoxicity [28]. Cell viability was assessed by incubating the cells with various concentrations of the probes (0, 10^−2^, 10^−4^, 10^−6^, and 10^−8^ M) for 24 h.

Trials have shown that the survival rate of *Vero* cells cultured with the fluorophores **9** and **10** for 24 h exceeds 90%, even at the highest concentration (10^−2^ M) of fluorophores. These findings demonstrate low biotoxicity and suggest that these compounds can be a practical tool for labeling cell organelles in complex biological environments (Figure 9). At the same time, phototoxicity was assessed using a box equipped with a UV lamp. No significant phototoxic effects were observed (Figure 9).

The significant TICT, large Stokes shift, and non-toxicity of the new fluorophores inspired us to evaluate their cell permeability and intracellular localization. The behavior of **9a**–**e** and **10a**–**e** in living cells using confocal laser scanning microscopy (CLSM). *Vero* cells were incubated with probes **9** and **10** for 0.5 h (working concentration of fluorophores 10^−5^ M). The investigated compounds were irradiated by lasers with wavelengths of 405, 458, and 488 nm. The emission spectra of the substances were extracted from the images obtained in lambda mode (Figure 10 and Figure 11). It is important to note that while a confocal microscope is a powerful imaging tool, it is not a spectrofluorometer, and the fluorescence spectra obtained may not be fully accurate. As illustrated in Figure 12 and Figure 13, all ten fluorophores successfully entered the cells and demonstrated good to excellent contrast in the confocal micrographs.

All tested fluorescent probes, including compounds **9a**–**e** and **10a**–**e**, exhibited fluorescence in the blue-green range of 406–481 nm. These compounds accumulated in lipid droplets and did not penetrate the cell nucleus (see Figure 10 and Figure 12). Additionally, most of the compounds, specifically **9b**, **9d**, **9e**, and **10c**–**e**, were also detected in the endoplasmic reticulum. Pyridine derivatives **9d** and **9e** were found in mitochondria as well. Notably, the most selective and promising fluorophores identified were **9a**,**c**, and **10a**,**b**.

Figure 10 illustrates that the pyridine derivatives **9a**,**c**–**e** exhibited bright fluorescence. In contrast, among the pyrimidine derivatives, the most significant fluorescence was demonstrated by compounds **10a**,**b**,**d**. It is important to note that the emission maxima for compounds **9b** and **10e**, which showed the brightest fluorescence, fall outside the detection range of the confocal microscope. Overall, these findings align well with the principles of aggregation-induced emission (Figure 5, Figure 6, Figure 7, Figure 8 and Appendix A), which may occur when a fluorescent probe DMSO solution is diluted in an aqueous cell medium.

It is interesting to note that, in general, a blue shift in the emission maxima was observed in cells compared to both the acetonitrile solution and the solid state (see Figure 10 and Table 1). This phenomenon may be attributed to the loss of planarity in push–pull systems due to the increasing viscosity of the lipid droplet medium. Additionally, it is known that lipid droplets have very low polarity, which contributes to the observed shorter wavelength emission [29].

Another interesting fact observed in this experiment is the variation of the fluorescence wavelength depending on the excitation wavelength (excitation-dependent emission effect) and its manifestation in different cellular compartments. In particular, for compounds **9c** and **9d**, with an increase in the excitation maxima from 405 to 488 nm, a red shift of the fluorescence maxima from 475 to 525 nm occurs (Figure 11). This phenomenon is known to be explained by complex relaxation processes in viscous lipid media [30].

To demonstrate the universality of lipid droplet staining, we conducted a series of experiments using compound **9a** on various cell cultures. We specifically tested *HaCaT*, *HEK-293t*, and *CaCo2*, which are commonly used model cell lines in biotechnology and medical research. Figure 14 illustrates the lipid droplet staining observed in these cultures. Although the staining images are not as clear-cut as those obtained with *Vero* cells due to differences in cell morphology, the staining patterns remain consistent across all cultures.

A colocalization study was conducted to demonstrate the selectivity of lipid droplet staining. To achieve this, cells were stained with the test substance alongside commercial dyes that are specific to different organelles. Figure 15 displays the staining results, showing the cells treated with both the test substance **9a** and the commercial dyes. The overlap of the staining appears to be nearly perfect, with a Manders coefficient of approximately 0.8.

Figure 16 displays the staining of cells with commercial dyes targeting organelles like lysosomes and mitochondria. No significant colocalization was detected with these dyes.

## 3. Experimental

Detailed specifications of the chemical substances used and methods for their characterizations are provided in the Appendix A.

**Synthesis of (*Z*)-1-[4-(dimethylamino)phenyl]-4,4,4-trifluoro-3-hydroxybut-2-en-1-one (3).** In a 250 mL round bottom flask, finely powdered lithium hydride (0.6 g, 75 mmol) was suspended in 100 mL of methyl *tert*-butyl ether (MTBE), and ethyl trifluoroacetate (**2**) (11.00 g, 77.5 mmol) was added. 4-Acetyl-*N*,*N*-dimethylaniline **1 **(8.16 g, 50 mmol) was then added in one portion with vigorous stirring and the reaction mixture was stirred at refluxing until TLC (silica, CHCl_3_) showed complete consumption of the starting material (ca. 7 h). All volatiles were removed on a rotary evaporator, a residue was dissolved in glacial acetic acid (ca. 20 mL), and 85% phosphoric acid (8.50 g, 73.7 mmol) was added. The obtained solution was diluted with water (ca. 200 mL), and a precipitate was filtered off, washed with water (3 × 50 mL), and dried on air to yield 11.80 g (91%) of** 3** as yellow powder. m.p. 69–71 °C. ^1^H NMR (500 MHz, CDCl_3_) δ 3.11 (s, 6H, 2CH_3_), 6.67–6.70 (m, 2H, H-3,5 Ar), 6.44 (s, 1H), 7.85–7.88 (m, 2H, H-2,6 Ar), 15.82 (br. s, 1H, enol-OH). ^19^F NMR (471 MHz, CDCl_3_) δ 85.61 (s, CF_3_). ^13^C NMR (126 MHz, CDCl_3_) δ 40.00 (s, (CH_3_)_2_N), 90.46 (q, *J* = 2.2 Hz, =CH-), 111.12, 117.73 (q, *J* = 282.6, CF_3_), 119.40, 130.12, 154.34, 174.47 (q, *J* = 35.4 Hz, CF_3_C=O), 186.05 (s, C=O). Calcd. for C_12_H_12_F_3_NO_2_ (259.23): C, 55.60; H, 4.67; N, 5.40, F, 21.99. Found: C, 55.43; H, 4.74; N, 5.22; F, 22.09.

**Synthesis of 6-[4-(dimethylamino)phenyl]-2-hydroxy-4-(trifluoromethyl) nicotino-nitrile (4).** A solution of diketone **3** (3.0 g, 11.57 mmol) and 2-cyanoacetamide (1.50 g, 17.86 mmol) in pyridine (15 mL) was kept on reflux for 20 h. Then, pyridine was distilled off until crystallization began. A residue diluted with 100 mL of water and acetic acid (ca. 5 mL) was added, a precipitate was filtered off, washed with water (3 × 40 mL), and dried on air. The obtained powder was washed with hot chloroform (3 × 20 mL) and re-precipitated from DMSO (10 mL) with water (100 mL). A precipitate was filtered off, washed with water (4 × 40 mL), and dried on air to yield 3.09 g (87%) of **4** as a deep-red powder. decomp. > 280 °C. ^1^H NMR (500 MHz, DMSO-*d*_6_) δ 3.05 (s, 6H, 2CH_3_), 6.78–6.80 (m, 2H, H-3,5 Ar), 6.98 (broad s, 1H, H-5 Py), 7.89–7.91 (broad m, 2H, H-2,6 Ar), 13.00 (s, 1H, OH). ^19^F NMR (471 MHz, DMSO-*d*_6_) δ 98.89 (CF_3_). ^13^C NMR (126 MHz, DMSO-*d*_6_) δ 98.29 (br. s, C-5 Py), 111.55 (s, C-2,6 Ar), 114.10 (s, CN), 116.44 (br. s, C-3 Py), 121.40 (q, ^1^*J* = 276.2 Hz, CF_3_), 129.41 (s, C-3,5 Ar), 144.87 (br. m, C-4 Py),152.75 (s, C-4 Ar), 154.83 (br. s, C-2 Py) 161.62 (br. s, C-6 Py), (CH_3_)_2_N—overlapped with DMSO. The signal of C-1 was not found because of broadening and low intensity. Calcd. for C_15_H_12_F_3_N_3_O (307.28): C, 58.63; H, 3.94; N, 13.68; F, 18.55. Found: C, 58.81; H, 4.13; N, 13.86; F, 18.41.

**Synthesis of 2-chloro-6-[4-(dimethylamino)phenyl]-4-(trifluoromethyl) nicotino-nitrile (5).** A suspension of **4** (2.0 g, 6.51 mmol) and tetraethylammonium chloride (1.1 g, 6.64 mmol) in POCl_3_ (10 g, 65.1 mmol) was stirred at ca. 80 °C for 2 h. The reaction mixture was stirred at reflux until complete consumption of the starting material (ca. 6 h). After cooling, the reaction mixture was poured into a mixture of ice (50 g) and water (100 mL) and neutralized carefully with solid NaHCO_3_. A precipitate was filtered off, washed with water (3 × 40 mL), dried on air, and dissolved in CH_2_Cl_2_ (ca. 15 mL). The obtained solution was passed through a silica pad (ca. 3 cm) and the silica was washed with CH_2_Cl_2_ (4 × 5 mL). Combined solutions were dissolved in methanol (ca. 50 mL), CH_2_Cl_2_ was distilled off on a rotary evaporator at standard pressure and the obtained suspension was cooled at 0–4 °C for 1 h. A precipitate was filtered off, washed with cold methanol (3 × 10 mL), and dried on air to yield **5** as a bright-red crystalline powder. The methanol solution was concentrated to give an additional crop of **5**. The total yield was 1.76 g (83%). m.p. 182–184 °C. ^1^H NMR (500 MHz, CDCl_3_) δ 3.10 (s, 6H, 2CH_3_), 6.72–6.76 (m, 2H, H-3,5Ar), 7.80 (s, 1H, H-5 Py), 7.98–8.02 (m, 2H, H-2,6 Ar). ^19^F NMR (471 MHz, CDCl_3_) δ 97.50 (s, CF_3_). ^13^C NMR (126 MHz, CDCl_3_) δ 40.03 (s, 2CH_3_), 100.72 (q, ^3^*J* = 1.6 Hz, C-3 Py), 111.76 (s, C-2,6 Ar), 112.51 (q, ^3^*J* = 4.3 Hz, C-5 Py), 112.80 (s, CN), 121.12 (q, ^1^*J* = 275.4 Hz, CF_3_), 121.64 (s, C-1 Ar), 129.43 (s, C-3,5 Ar), 142.60 (q, ^2^*J* = 33.7 Hz, C-4 Py),153.01 (s, C-4 Ar), 154.54 (s, C-2 Py), 161.25 (s, C-6 Py). Calcd. for C_15_H_11_ClF_3_N_3_ (325.72): C, 55.31; H, 3.40; N, 12.90; F, 17.50. Found: C, 55.21; H, 3.22; N, 12.89; F, 17.58.

**Synthesis of 4-[4-(dimethylamino)phenyl]-6-(trifluoromethyl)pyrimidin-2-ol (6).** A solution of diketone **3** (3.0 g, 11.57 mmol), urea (1.4 g, 23.33 mmol), and triethylborate (6.8 g, 46.57 mmol) in acetonitrile (20 mL) was refluxed for 20 h. Then, all volatiles were removed, and a residue was washed several times with water (4 × 20 mL), dried on air, and crystallized from chloroform to yield **6** as an orange-yellow crystalline powder. The chloroform solution was concentrated to give an additional crop of **6**. The total yield was 2.88 g (88%). m.p. 260–264 °C (sublim.). ^1^H NMR (500 MHz, DMSO-*d*_6_) δ 3.04 (s, 6H, 2CH_3_), 6.78–6.81 (m, 2H, H-3,5 Ar), 7.41 (broad s, 1H, H-5 Pyr), 8.05–8.08 (broad s, 2H, H-2,6 Ar), 12.40 (s, 1H, OH). ^19^F NMR (471 MHz, DMSO-*d*_6_) δ 93.14 (broad s, CF_3_). ^13^C NMR (126 MHz, DMSO-*d*_6_) δ 95.35 (s, C-5 Pyr), 111.43 (s, C-2,6 Ar), 120.43 (q, ^1^*J* = 276.4 Hz, CF_3_), 129.09 (s, C-3,5 Ar), 153.02 (s, C-4 Ar), (CH_3_)_2_N—overlapped with DMSO. Other signals are broad with low intensity. Calcd. for C_13_H_12_F_3_N_3_O (283.25): 55.12; H, 4.27; N, 14.84; F, 20.12. Found: C, 55.05; H, 4.43; N, 14.97; F, 20.06.

**Synthesis of 4-[2-chloro-6-(trifluoromethyl)pyrimidin-4-yl]-*N,N*-dimethylaniline (7).** A suspension of **7** (2.0 g, 7.06 mmol) and tetraethylammonium chloride (1.2 g, 7.25 mmol) in POCl_3_ (10 g, 65.1 mmol) was stirred at ca. 80 °C for 1 h. The reaction mixture was stirred at reflux until complete consumption of the starting material (ca. 6 h). After cooling, the reaction mixture was poured into a mixture of ice (50 g) and water (100 mL) and neutralized carefully with solid NaHCO_3_. A precipitate was filtered off, washed with water (3 × 40 mL), dried on air, and dissolved in CH_2_Cl_2_ (ca. 10 mL). The obtained solution was passed through a silica pad (ca. 2 cm) and the silica was washed with CH_2_Cl_2_ (3 × 5 mL). Combined solutions were dissolved in methanol (ca. 50 mL), CH_2_Cl_2_ was distilled off on a rotary evaporator at standard pressure and the obtained suspension was cooled at 0–4 °C for 1 h. A precipitate was filtered off, washed with cold methanol (3 × 10 mL), and dried on air to yield **7** as bright-yellow crystalline powder. The methanol solution was concentrated to give an additional crop of **7**. The total yield was 1.68 g (79%). m.p. 151–152 °C. ^1^H NMR (500 MHz, CDCl_3_) δ 3.11 (s, 6H, 2CH_3_), 6.72–6.76 (m, 2H, H-3,5 Ar), 7.75 (s, 1H, H-5 Pyr), 8.04–8.08 (m, 2H, H-2,6 Ar). ^19^F NMR (471 MHz, CDCl_3_) δ 91.75 (s, CF_3_).^13^C NMR (126 MHz, CDCl_3_) δ 40.02 (s, 2CH_3_), 108.70 (q, ^3^*J* = 2.8 Hz, C-5 Pyr), 111.61 (s, C-2,6 Ar), 120.19 (q, ^1^*J* = 275.1 Hz, CF_3_), 120.70 (s, C-1 Ar), 129.42 (s, C-3,5 Ar), 153.52 (s, C-4 Ar), 157.07 (q, ^2^*J* = 36.1 Hz, C-6 Pyr), 161.98 (s, C-2 Pyr), 168.94 (s, C-4 Pyr). Calcd. for C_13_H_11_ClF_3_N_3_ (301.70): C, 51.75; H, 3.68; N, 13.93; F, 18.89. Found: C, 51.78; H, 3.48; N, 13.96; F, 18.93.

**General procedure for the Suzuki cross-coupling reactions exploited for the synthesis of compounds 9 and 10:** A mixture of the 2-chloro-6-[4-(dimethylamino)phenyl]-4-(trifluoromethyl)nicotinonitrile (**5**) (163 mg, 0.5 mmol) [or 4-[2-chloro-6-(trifluoromethyl) pyrimidin-4-yl]-*N*,*N*-dimethylaniline (**7**) (151 mg, 0.5 mmol), (het)arylboronic acid or the corresponding pinacol ester (0.6 mmol, 1.2 equiv.), Pd(PPh_3_)_4_ (29 mg, 5 mol %), and K_3_PO_4_ (265 mg, 1.25 mmol) were dissolved in 1,4-dioxane 10 mL. The reaction was refluxed under an argon atmosphere for 20 h. The reaction mixture was cooled to room temperature, the formed suspension was filtered through a small plug of SiO_2_ (3–4 cm), which was successively washed twice with 1,4-dioxane (2 × 5 mL), and the obtained filtrate was evaporated under reduced pressure to dryness. The resulting residue was purified by column chromatography (SiO_2_; EtOAc/hexane 1:8) and further crystallized from MeOH to afford the desired cross-coupling products **9** [or **10**].

**6-[4-(Dimethylamino)phenyl]-2-(4-(diphenylamino)phenyl)-4-(trifluoromethyl) nicotinonitrile (9a)**. Yield 147 mg (55%), a yellow-orange solid, mp 171–173 °C. ^1^H NMR (600 MHz, DMSO-*d*_6_) δ 8.24–8.12 (m, 3H), 7.87 (d, *J* = 8.3 Hz, 2H), 7.40 (t, *J* = 7.6 Hz, 4H), 7.16 (d, *J* = 8.3 Hz, 6H), 7.03 (d, *J* = 8.4 Hz, 2H), 6.80 (d, *J* = 8.6 Hz, 2H), 3.04 (s, 6H). ^19^F NMR (376 MHz, DMSO-*d*_6_) δ 99.81 (s, CF_3_). ^13^C NMR (151 MHz, DMSO-*d*_6_) δ 161.1 (d, *J* = 252.8 Hz), 152.9, 149.9, 146.9, 141.0 (q, *J* = 32.2 Hz), 131.0, 130.3, 129.9, 129.8, 125.88, 125.2, 124.9, 123.0, 122.5 (d, *J* = 274.9 Hz), 120.6, 119.7, 112.8 (d, *J* = 4.1 Hz), 112.2, 96.7, 25.6. HRMS (ESI): *m*/*z* calcd for C_33_H_26_F_3_N_4_: 535.2104 [M+H]^+^; found: 535.2096 and *m*/*z* calcd for C_33_H_25_F_3_N_4_Na: 557.1924 [M+Na]^+^; found: 557.1921.

**2-[4-(9*H*-Carbazol-9-yl)phenyl]-6-(4-(dimethylamino)phenyl)-4-(trifluoromethyl) nicotinonitrile (9b)**. Yield 133 mg (50%), a yellow-orange solid, mp 220–222 °C. ^1^H NMR (600 MHz, DMSO-*d*_6_) δ 8.35–8.25 (m, 7H), 7.89 (d, *J* = 8.0 Hz, 2H), 7.53 (d, *J* = 8.2 Hz, 2H), 7.48 (t, *J* = 7.6 Hz, 2H), 7.34 (t, *J* = 7.4 Hz, 2H), 6.84 (d, *J* = 8.6 Hz, 2H), 3.05 (s, 6H). ^19^F NMR (376 MHz, DMSO-*d*_6_) δ 97.71 (s, CF_3_). ^13^C NMR (151 MHz, DMSO-*d*_6_) δ 161.8, 160.5, 153.1, 141.0 (q, *J* = 32.5 Hz), 140.3, 139.3, 136.3, 130.8 (d, *J* = 252.0 Hz), 126.9, 126.9, 123.5, 122.9, 122.5 (d, *J* = 275.6 Hz), 121.1, 120.9, 116.0, 113.77, 113.75, 112.2, 110.2, 97.9. (CH_3_)_2_N—overlapped with DMSO. HRMS (ESI): *m*/*z* calcd for C_33_H_24_F_3_N_4_: 533.1948 [M+H]^+^; found: 533.1947 and *m*/*z* calcd for C_33_H_23_F_3_N_4_Na: 555.1767 [M+Na]^+^; found: 555.1764.

**6-[4-(Dimethylamino)phenyl]-2-(9-ethyl-9*H*-carbazol-3-yl)-4-(trifluoromethyl) nicotinonitrile (9c)**. Yield 124 mg (51%), a yellow-orange solid, mp 179–181 °C. ^1^H NMR (600 MHz, DMSO-*d*_6_) δ 8.78 (s, 1H), 8.35–8.19 (m, 4H), 8.06 (d, *J* = 8.5 Hz, 1H), 7.81 (d, *J* = 8.5 Hz, 1H), 7.70 (d, *J* = 8.2 Hz, 1H), 7.54 (t, *J* = 7.7 Hz, 1H), 7.28 (t, *J* = 7.5 Hz, 1H), 6.84 (d, *J* = 8.5 Hz, 2H), 4.53 (q, *J* = 7.2 Hz, 2H), 3.04 (s, 6H), 1.38 (t, *J* = 7.2 Hz, 3H). ^19^F NMR (376 MHz, DMSO-*d*_6_) δ 99.90 (s, CF_3_). ^13^C NMR (151 MHz, DMSO-*d*_6_) δ 163.6, 160.3, 152.9, 141.05, 141.04 (d, *J* = 31.9 Hz), 140.6, 129.9, 128.3, 127.5, 126.9, 123.5, 123.2, 122.6 (d, *J* = 27.2 Hz), 122.3, 121.6, 121.2, 119.9, 116.5, 112.7, 112.2, 110.0, 109.6, 97.4, 40.5, 37.7, 14.2. HRMS (ESI): *m*/*z* calcd for C_29_H_24_F_3_N_4_: 485.1948 [M+H]^+^; found: 485.1941 and *m*/*z* calcd for C_29_H_23_F_3_N_4_Na: 507.1767 [M+Na]^+^; found: 507.1764.

**6-[4-(Dimethylamino)phenyl]-2-(4-fluorophenyl)-4-(trifluoromethyl)nicotinonitrile (9d)**. Yield 94 mg (49%), a yellow-orange solid, mp 168–170 °C. ^1^H NMR (600 MHz, DMSO-*d*_6_) δ 8.26 (s, 1H), 8.20 (d, *J* = 8.6 Hz, 2H), 8.06–7.91 (m, 2H), 7.45 (t, *J* = 8.7 Hz, 2H), 6.81 (d, *J* = 8.6 Hz, 2H), 3.04 (s, 6H). ^19^F NMR (376 MHz, DMSO-*d*_6_) δ 99.84 (s, CF_3_), 52.15 (tt, *J* = 8.9, 5.4 Hz, 1F). ^13^C NMR (151 MHz, DMSO-*d*_6_) δ 163.8 (d, *J* = 248.5 Hz), 161.7, 160.4, 153.0, 140.8 (q, *J* = 32.2 Hz), 134.0 (d, *J* = 3.1 Hz), 132.1 (d, *J* = 8.8 Hz), 129.8, 122.8, 122.4 (q, *J* = 275.1 Hz), 116.0 (d, *J* = 21.9 Hz), 115.9, 113.6 (d, *J* = 4.6 Hz), 112.2, 97.9, (CH_3_)_2_N—overlapped with DMSO. HRMS (ESI): *m*/*z* calcd for C_21_H_16_F_4_N_3_: 386.1275 [M+H]^+^; found: 386.1267 and *m*/*z* calcd for C_21_H_15_F_4_N_3_Na: 408.1094 [M+Na]^+^; found: 408.1092.

**6-[4-(Dimethylamino)phenyl]-4-(trifluoromethyl)-2-[4-(trifluoromethyl)phenyl] nicotinonitrile (9e)**. Yield 113 mg (52%), a yellow-orange solid, mp 244–246 °C. ^1^H NMR (600 MHz, DMSO-*d*_6_) δ 8.36 (s, 1H), 8.24 (d, *J* = 8.6 Hz, 2H), 8.15 (d, *J* = 7.9 Hz, 2H), 8.00 (d, *J* = 8.0 Hz, 2H), 6.83 (d, *J* = 8.7 Hz, 2H), 3.06 (s, 6H). ^19^F NMR (376 MHz, DMSO-*d*_6_) δ 101.32 (s, CF_3_), 99.90 (s, CF_3_). ^13^C NMR (151 MHz, DMSO-*d*_6_) δ 161.5, 160.6, 153.1, 141.5, 140.8 (q, *J* = 32.7 Hz), 130.8 (q, *J* = 31.8 Hz), 130.6, 130.0, 126.0 (d, *J* = 3.0 Hz), 124.5 (d, *J* = 272.4 Hz), 122.7, 122.4 (d, *J* = 275.5 Hz), 115.6, 114.2, 112.2, 98.5, (CH_3_)_2_N—overlapped with DMSO. HRMS (ESI): *m*/*z* calcd for C_22_H_16_F_6_N_3_: 436.1243 [M+H]^+^; found: 436.1233 and *m*/*z* calcd for C_22_H_15_F_6_N_3_Na: 458.1062 [M+Na]^+^; found: 458.1062.

**4-{4-[4-(Dimethylamino)phenyl]-6-(trifluoromethyl)pyrimidin-2-yl}-*N,N*-diphenyl aniline (10a)**. Yield 176 mg (69%), a yellow solid, mp 145–146 °C. ^1^H NMR (500 MHz, CDCl_3_) δ 8.47–8.41 (m, 2H), 8.20–8.14 (m, 2H), 7.68 (s, 1H), 7.32–7.27 (m, 4H), 7.19–7.12 (m, 6H), 7.11–7.06 (m, 2H), 6.81–6.75 (m, 2H), 3.09 (s, 6H). ^19^F NMR (471 MHz, CDCl_3_) δ 91.50 (s, CF_3_). ^13^C NMR (126 MHz, CDCl_3_) δ 165.7, 164.7, 155.7 (q, *J* = 35.0 Hz), 152.8, 150.5, 147.2, 130.4, 129.7, 129.3, 128.7, 125.2, 123.6, 123.1, 121.9, 120.1 (q, *J* = 275.1 Hz), 111.7, 107.2 (d, *J* = 3.0 Hz), 40.1. Calcd. for C_31_H_25_F_3_N_4_ (510.56): C, 72.93; H, 4.94; N, 10.97. Found: C, 72.90; H, 4.91; N, 10.99.

**4-{2-[4-(9*H*-Carbazol-9-yl)phenyl]-6-(trifluoromethyl)pyrimidin-4-yl}-*N,N*-dimethyl aniline (10b)**. Yield 226 mg (89%), a yellow solid, mp 190–191 °C. ^1^H NMR (500 MHz, CDCl_3_) δ 8.88–8.82 (m, 2H), 8.27–8.21 (m, 2H), 8.19–8.13 (m, 2H), 7.81 (s, 1H), 7.78–7.72 (m, 2H), 7.55–7.50 (m, 2H), 7.48–7.40 (m, 2H), 7.35–7.28 (m, 2H), 6.85–6.80 (m, 2H), 3.12 (s, 6H). ^19^F NMR (471 MHz, CDCl_3_) δ 91.60. (s, CF_3_). ^13^C NMR (126 MHz, CDCl_3_) δ 166.1, 164.2, 155.9 (d, *J* = 35.1 Hz), 152.9, 140.5, 140.3, 136.0, 130.2, 128.9, 126.7, 126.0, 123.6, 122.7, 121.1 (d, *J* = 275.3 Hz), 120.3, 120.2, 111.7, 109.9, 108.2 (d, *J* = 3.0 Hz), 40.1. Calcd. for C_31_H_23_F_3_N_4_ (508.55): C, 73.22; H, 4.56; N, 11.02. Found: C, 73.19; H, 4.53; N, 11.04.

**4-[2-(9-Ethyl-9*H*-carbazol-3-yl)-6-(trifluoromethyl)pyrimidin-4-yl]-*N,N*-dimethyl aniline (10c)**. Yield 170 mg (74%), a beige solid, mp 161–162 °C. ^1^H NMR (400 MHz, CDCl_3_) δ 9.37 (d, *J* = 1.6 Hz, 1H), 8.78 (dd, *J* = 8.7, 1.7 Hz, 1H), 8.32–8.21 (m, 3H), 7.71 (s, 1H), 7.55–7.46 (m, 2H), 7.48–7.41 (m, 1H), 7.34–7.25 (m, 1H), 6.88–6.81 (m, 2H), 4.43 (q, *J* = 7.2 Hz, 2H), 3.11 (s, 6H), 1.49 (t, *J* = 7.2 Hz, 3H). ^19^F NMR (471 MHz, CDCl_3_) δ 91.61(s, CF_3_). ^13^C NMR (126 MHz, CDCl_3_) δ 165.81, 165.78, 155.8 (q, *J* = 35.1 Hz), 152.7, 142.0, 140.5, 128.8, 128.2, 126.6, 125.9, 123.5, 123.3, 123.2, 121.5, 121.3 (q, *J* = 275.1 Hz), 120.9, 119.4, 111.7, 108.7, 108.2, 107.1 (d, *J* = 2.7 Hz), 40.1, 37.7, 13.8. Calcd. for C_27_H_23_F_3_N_4_ (460.50): C, 70.42; H, 5.03; N, 12.17. Found: C, 70.41; H, 5.02; N, 12.20.

**4-[2-(4-Fluorophenyl)-6-(trifluoromethyl)pyrimidin-4-yl]-*N,N*-dimethylaniline (10d)**. Yield 128 mg (71%), a yellow solid, mp 145–146 °C. ^1^H NMR (400 MHz, CDCl_3_) δ 8.67–8.57 (m, 2H), 8.23–8.14 (m, 2H), 7.74 (s, 1H), 7.24–7.14 (m, 2H), 6.85–6.78 (m, 2H), 3.10 (s, 6H). ^19^F NMR (471 MHz, CDCl_3_) δ 91.53 (s, CF_3_), 52.60–51.36 (m, 1F). ^13^C NMR (126 MHz, CDCl_3_) δ 166.0, 165.0 (d, *J* = 250.6 Hz), 164.0, 155.8 (q, *J* = 35.3 Hz), 152.9, 133.3 (d, *J* = 2.6 Hz), 130.8 (d, *J* = 8.6 Hz), 128.8, 122.8, 121.1 (q, *J* = 275.1 Hz), 115.4 (d, *J* = 21.5 Hz), 111.7, 107.9 (d, *J* = 3.0 Hz), 40.1. Calcd. for C_19_H_15_F_4_N_3_ (361.34): C, 63.16; H, 4.18; N, 11.63. Found: C, 63.17; H, 4.15; N, 11.60.

***N,N*-Dimethyl-4-{6-(trifluoromethyl)-2-[4-(trifluoromethyl)phenyl]pyrimidin-4-yl} aniline (10e)**. Yield 154 mg (75%), an orange solid, mp 169–170 °C. ^1^H NMR (400 MHz, CDCl_3_) δ 8.76–8.68 (m, 2H), 8.24–8.16 (m, 2H), 7.81 (s, 1H), 7.80–7.73 (m, 2H), 6.86–6.79 (m, 2H), 3.11 (s, 6H). ^19^F NMR (376 MHz, CDCl_3_) δ 98.98 (s, CF_3_), 91.60 (s, CF_3_). ^13^C NMR (126 MHz, CDCl_3_) δ 166.2, 163.6, 155.9 (q, *J* = 35.1 Hz), 153.0, 140.4, 132.6 (q, *J* = 32.3 Hz), 128.88, 128.86, 125.4 (q, *J* = 3.8 Hz), 124.1 (q, *J* = 272.5 Hz), 122.5, 121.0 (q, *J* = 275.3 Hz), 111.7, 108.7 (d, *J* = 3.1 Hz), 40.1. Calcd. for C_20_H_15_F_6_N_3_ (411.35): C, 58.40; H, 3.68; N, 10.22. Found: C, 58.38; H, 3.66; N, 10.23.

## 4. Conclusions

In summary, we designed and developed ten novel AIE fluorescent probes based on CF_3_-substituted pyridine and pyrimidine with a donor-acceptor-donor (D-A-D) structure, specifically for imaging lipid droplets. We have thoroughly investigated the photophysical properties of these compounds in both solution and solid state. Most of the fluorophores exhibited large Stokes shifts and the aggregation-induced emission effect, which can be explained by the sterically hindering aromatic substituent at the C(2) position. These fluorophores are biologically available and can easily penetrate living cells, accumulating in lipid droplets. Comparison with known dyes and studies on various cell cultures demonstrated the selectivity of the obtained push–pull systems for visualizing lipid droplets. We believe that pyridine and pyrimidine derivatives hold great potential as versatile scaffolds for the development of fluorescent probes for bioimaging.

## Data Availability

The original research data is available upon request.

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
