# Peer review of "Novel CF₃-Substituted Pyridine- and Pyrimidine-Based Fluorescent Probes for Lipid Droplet Bioimaging"

_ijms, 2025, doi:10.3390/ijms26115271_

Round 1
Reviewer 1 Report
Comments and Suggestions for Authors
- Scheme 2: Could the authors explain why they did not investigate the reaction time to improve yields?
- Citation Formatting: The citation numbers are not in a standard format and should be revised for consistency.
- Title Revision: In my opinion, the current title does not fully reflect the content of the article, particularly the role of fluorinated substituents. I propose amending it to:
"Novel CF₃-Substituted Pyridine- and Pyrimidine-Based Fluorescent Probes for Lipid Droplet Bioimaging."
Author Response
Dear Editor,
Thank you for your letter and for the reviewers’ comments concerning our manuscript entitled " Novel CF₃-Substituted Pyridine- and Pyrimidine-based Fluorescent Probes for Lipid Droplets Bioimaging ". Those comments are all valuable and very helpful for revising and improving our paper, as well as the important guiding significance to our research. The manuscript has been carefully revised according to reviewers’ comments. These changes are shown as yellow highlights in the revised paper. We hope the revised manuscript would be suitable for publication in International Journal of Molecular Sciences.
Yours sincerely,
Authors
Reviewer 1.
Comments and Suggestions for Authors
- Scheme 2: Could the authors explain why they did not investigate the reaction time to improve yields?
Response:
We thank the reviewer for his/her attention. The authors did not focus on optimizing the conditions for the Suzuki cross-coupling reaction. The synthesis of the target compounds 9 and 10, as shown in Scheme 2, was carried out following the procedure we previously described in reference 22.
- Citation Formatting: The citation numbers are not in a standard format and should be revised for consistency.
Response:
We thank the reviewer for his/her attention. The references have been updated to meet the journal's guidelines.
- Title Revision: In my opinion, the current title does not fully reflect the content of the article, particularly the role of fluorinated substituents. I propose amending it to:
"Novel CF₃-Substituted Pyridine- and Pyrimidine-Based Fluorescent Probes for Lipid Droplet Bioimaging."
Response:
We thank the reviewer for his/her attention. The authors definitely agree with the reviewer's proposal. The title of the manuscript has been changed.

Reviewer 2 Report
Comments and Suggestions for Authors
In the submitted manuscript, authors devised fluorescent probes for lipid droplets bioimaging. Some issues should be resolved before accepting, as following:
- Fluorescence intensity changes of the molecule were only tested under different MeCN/Hâ‚‚O ratios. For this reviewer think the following experiments further confirm its AIE properties: Conduct Transmission Electron Microscopy (TEM) to directly observe morphological aggregation state changes; Perform Dynamic Light Scattering (DLS) to measure nanoparticle size variations.
- Perform co-localization experiments with commercial lipid droplet dye BODIPY to verify targeting efficacy through fluorescence signal overlap.
- Co-incubate with other organelle dyes (e.g., mitochondria, lysosomes, endoplasmic reticulum, nucleus) and observe fluorescence signal overlap to confirm specificity.
- Perform experiments on multiple types of cells.
-
The article mentioned good cell viability. Could aurthors add the following experiments further verify its biosafety: Conduct phototoxicity tests using the MTT assay to validate cell viability; Evaluate its photostability.
- Some recent related research or review papers should be cited: smart molecules, 2024, 2, e20240040; Chemical & Biomedical Imaging, 2025, doi: 10.1021/cbmi.4c00096.
Author Response
Dear Editor,
Thank you for your letter and for the reviewers’ comments concerning our manuscript entitled " Novel CF₃-Substituted Pyridine- and Pyrimidine-based Fluorescent Probes for Lipid Droplets Bioimaging". Those comments are all valuable and very helpful for revising and improving our paper, as well as the important guiding significance to our research. The manuscript has been carefully revised according to reviewers’ comments. These changes are shown as yellow highlights in the revised paper. We hope the revised manuscript would be suitable for publication in International Journal of Molecular Sciences.
Yours sincerely,
Authors
Reviewer 3.
Comments and Suggestions for Authors
In the submitted manuscript, authors devised fluorescent probes for lipid droplets bioimaging. Some issues should be resolved before accepting, as following:
- Fluorescence intensity changes of the molecule were only tested under different MeCN/Hâ‚‚O ratios. For this reviewer think the following experiments further confirm its AIE properties: Conduct Transmission Electron Microscopy (TEM) to directly observe morphological aggregation state changes; Perform Dynamic Light Scattering (DLS) to measure nanoparticle size variations.
Response:
We thank the reviewer for his/her attention. Unfortunately, the authors currently lack the necessary equipment to conduct Transmission Electron Microscopy (TEM) and Dynamic Light Scattering (DLS). As a result, we are unable to implement the reviewer's suggestion at this time. However, we will consider this suggestion in future studies concerning the AIE properties of our compounds.
- Perform co-localization experiments with commercial lipid droplet dye BODIPY to verify targeting efficacy through fluorescence signal overlap.
Response:
We thank the reviewer for his/her attention. We utilized a commercially available dye BDP 650/665 (https://www.lumiprobe.com/p/bdp-650-665-lipid-stain) to label lipid droplets, demonstrating its colocalization with the substances tested. The results were added to the manuscript.
- Co-incubate with other organelle dyes (e.g., mitochondria, lysosomes, endoplasmic reticulum, nucleus) and observe fluorescence signal overlap to confirm specificity.
Response:
We thank the reviewer for his/her attention. We conducted colocalization tests using dyes for lysosomes and mitochondria. Unfortunately, we currently do not have access to suitable dyes for labeling the cell nucleus or endoplasmic reticulum. Additionally, it is visually evident that there is no colocalization in the nucleus when we overlap the transmission channel with the fluorescent channel. The relevant results have been included in the article.
- Perform experiments on multiple types of cells.
Response:
We thank the reviewer for his/her attention. We conducted experiments on HaCaT, HEK-293T, and CaCo2 cell cultures. The results from these cultures commonly repeated those observed in Vero culture. The relevant data have been included in the article.
- The article mentioned good cell viability. Could authors add the following experiments further verify its biosafety: Conduct phototoxicity tests using the MTT assay to validate cell viability; Evaluate its photostability.
Response:
We thank the reviewer for his/her attention. We conducted a phototoxicity test on substances 9a-e and 10a-e by assessing their viability using a resazurin test, which revealed no significant toxicity. Additionally, confocal microscope observations did not indicate any notable phototoxicity in situ. The relevant data has been included in the article.
- Some recent related research or review papers should be cited: smart molecules, 2024, 2, e20240040; Chemical & Biomedical Imaging, 2025, doi: 10.1021/cbmi.4c00096.
Response:
We thank the reviewer for his/her attention. The relevant references have been included in the article (see references 12 and 13).
The English has been checked and improved.

Reviewer 3 Report
Comments and Suggestions for Authors
The article presents fascinating synthetic research, but the method of its preparation leaves much to be desired.
The article was prepared very sloppily. There are no literature references in the text. No Supporting Information attached, and the authors refer to it in the text. Table 1 with the results of photophysical studies requires reformatting to make it more readable.
Please take some time to prepare it carefully and then please submit it again.
Author Response
The article presents fascinating synthetic research, but the method of its preparation leaves much to be desired.
The article was prepared very sloppily. There are no literature references in the text. No Supporting Information attached, and the authors refer to it in the text. Table 1 with the results of photophysical studies requires reformatting to make it more readable.
Please take some time to prepare it carefully and then please submit it again.
Response:
We thank the reviewer for his/her attention. The authors apologize for the misunderstanding that arose when transferring the article to the publishing system. All references were carefully added to the text of the manuscript. Table 1 was edited. The file with Supporting Information was re-uploaded.

Round 2
Reviewer 2 Report
Comments and Suggestions for Authors
Authors have revised manuscript ccarefully based on the reviewers' comments. So, it can be accepted.
Reviewer 3 Report
Comments and Suggestions for Authors
The synthesis of new fluorescent derivatives of pyridines and pyrimidines was designed and performed. Their photophysical properties were investigated through absorption and emission spectroscopy in acetonitrile solutions and the solid state. Chemical structures were confirmed using standard methods (1H, 13C, and 19F NMR). It was shown that the obtained compounds can be successfully used for fluorescent visualization of lipid droplets, as they are non-toxic and show AIE-induced fluorescence.
In the discussion of the results, there was no comparison of the advantages and disadvantages of the obtained compounds with the already known compounds used to visualize lipid droplets.
The authors write in the text that the structure was confirmed by performing HRMS spectra, but no such spectra were included in the SI. If such spectra have been recorded, they should be included in the SI.
What is the purpose of the yellow ellipse in Figure 1?
Author Response
Dear Editor,
Thank you for your letter and for the reviewers’ comments concerning our manuscript entitled " Novel CF₃-Substituted Pyridine- and Pyrimidine-based Fluorescent Probes for Lipid Droplets Bioimaging". Those comments are all valuable and very helpful for revising and improving our paper, as well as the important guiding significance to our research. The manuscript has been carefully revised according to reviewers’ comments. These changes are shown as yellow highlights in the revised paper. We hope the revised manuscript would be suitable for publication in International Journal of Molecular Sciences.
Yours sincerely,
Authors
Reviewer 4.
Comments and Suggestions for Authors
The synthesis of new fluorescent derivatives of pyridines and pyrimidines was designed and performed. Their photophysical properties were investigated through absorption and emission spectroscopy in acetonitrile solutions and the solid state. Chemical structures were confirmed using standard methods (1H, 13C, and 19F NMR). It was shown that the obtained compounds can be successfully used for fluorescent visualization of lipid droplets, as they are non-toxic and show AIE-induced fluorescence.
In the discussion of the results, there was no comparison of the advantages and disadvantages of the obtained compounds with the already known compounds used to visualize lipid droplets.
Response:
We thank the reviewer for his/her attention. The additional section comparing compound 9a with known lipid droplet visualization probes has been added to the text.
The authors write in the text that the structure was confirmed by performing HRMS spectra, but no such spectra were included in the SI. If such spectra have been recorded, they should be included in the SI.
Response:
We thank the reviewer for his/her attention. HRMS spectra for compounds 9a-e were included in the Supporting Information file as figures S93-S97. HRMS spectra for all other compounds were not performed.
What is the purpose of the yellow ellipse in Figure 1?
Response:
We thank the reviewer for his/her attention. The purpose of the yellow ellipse in Figure 1 is to draw readers' attention to the C-CN fragment (for compound I) and the N atom (for compound II).
